# Mechanical and Immunological Regulation in Wound Healing and Skin Reconstruction

**DOI:** 10.3390/ijms22115474

**Published:** 2021-05-22

**Authors:** Shun Kimura, Takashi Tsuji

**Affiliations:** 1Laboratory for Organ Regeneration, RIKEN Centre for Biosystem Dynamics Research, Hyogo 650-0047, Japan; skimura@rohto.co.jp; 2ROHTO Pharmaceutical Co., Ltd., Osaka 544-8666, Japan

**Keywords:** wound healing, mechano-biology, tensional homeostasis, immunological interaction, skin equivalents

## Abstract

In the past decade, a new frontier in scarless wound healing has arisen because of significant advances in the field of wound healing realised by incorporating emerging concepts from mechanobiology and immunology. The complete integumentary organ system (IOS) regeneration and scarless wound healing mechanism, which occurs in specific species, body sites and developmental stages, clearly shows that mechanical stress signals and immune responses play important roles in determining the wound healing mode. Advances in tissue engineering technology have led to the production of novel human skin equivalents and organoids that reproduce cell–cell interactions with tissue-scale tensional homeostasis, and enable us to evaluate skin tissue morphology, functionality, drug response and wound healing. This breakthrough in tissue engineering has the potential to accelerate the understanding of wound healing control mechanisms through complex mechanobiological and immunological interactions. In this review, we present an overview of recent studies of biomechanical and immunological wound healing and tissue remodelling mechanisms through comparisons of species- and developmental stage-dependent wound healing mechanisms. We also discuss the possibility of elucidating the control mechanism of wound healing involving mechanobiological and immunological interaction by using next-generation human skin equivalents.

## 1. Introduction

The integumentary organ system (IOS), including skin and skin appendages such as hair, sebaceous glands, sweat glands, feathers, and nails, plays an essential role in protecting deeper tissues from extrinsic stress, such as dryness, chemical compounds, ultraviolet, and mechanical stress [1]. These external stresses cause not only microscale damage, such as protein denaturation and degradation, cellular senescence, apoptosis and abnormal differentiation, but also macroscopic tissue damage, such as wounds and fibrosis. To maintain the structural and functional homeostasis of the IOS, molecular-, cellular- and tissue-scale damage is repaired by the tissue reconstruction response. An incomplete tissue reconstruction response causes skin dysfunctions and dysmorphologies such as ulceration, fibrosis, pigmentation, wrinkles and sagging. Wound healing is the most dramatic tissue reconstruction response to skin tissue damage. Skin wound therapy is divided into two classes as “conventional” and “regenerative”. Conventional therapy often leads to scar formation and results in aesthetic and functional problems [2]. In research and development of “Regenerative wound therapy”, the complete regeneration of the IOS, including skin appendages, and avoidance of pathological wound healing such as hypertrophic scarring (HTS) and keloid formation are critical issues in the field of dermatology [3].

Comparative analysis of different wound healing mechanisms depending on species, body sites, and developmental stages is thought to provide a breakthrough in regenerative wound therapy [4]. Studies of amphibians and teleosts, which are model organisms with the ability to fully regenerate the IOS, suggest that mechanical stress due to the contraction of the wound site may inhibit complete wound regeneration in mammals [5,6,7,8,9,10,11]. In addition, a comparative analysis of the human wound healing process in adults and foetuses shows that a weak inflammatory response by undifferentiated foetal immune cells plays an important role in complete foetal IOS regeneration [12]. Interestingly, over the last decade, it has been found that complete skin regeneration is induced in mammals with extremely soft skin, and in large full-thickness wounds with sufficiently low tissue-scale tension [13,14,15]. Studies of the complete skin regeneration mechanism in adult mammals suggest that the interaction of mechanical stress and immune responses may determine the mode of skin regeneration, via differentiation and functional control of wound healing-related cells [4]. Wound healing, which is affected by spatiotemporal cell–cell interactions and mechanobiological control, is an extremely complex physiological response, and it has been difficult to elucidate its detailed mechanisms under in vivo conditions. However, recent advances in tissue engineering technology have improved the reproduction of tissue structure and function in human skin equivalents, enabling in vitro analysis of partial wound healing mechanisms [16,17,18].

In this review, we provide an overview of the control mechanism of wound healing induced by mechanical stress and the immune response, and discuss the possibility of wound treatment strategies that do not cause HTS or keloid formations.

## 2. Mechanical Regulation in Wound Healing

### 2.1. Mechanical Stress Is Involved in the Moderation of Wound Healing through the Regulation of Myofibroblast Differentiation

A comparative analysis of wound healing patterns in organisms with different skin mechanical properties shows that mechanical stress is associated with the wound healing mode. In general, wound healing in *Mus musculus* and *Homo sapiens* does not regenerate skin appendages including hair follicles, and the mechanical properties of the wound healing site change to those of intact skin. In particular, pathological wound healing, such as HTS and keloids, often occurs in *Homo sapiens*, who have higher skin tensile strength than *Mus musculus* (Figure 1a) [9,19,20]. It was demonstrated that the African spiny mouse (*Acomys*), which has extremely weak tensile strength compared to *Mus musculus* and *Homo sapiens*, has the potential to regenerate up to 70% of its skin, including all appendages, after full-thickness skin wounding [13]. A comparative analysis of the mechanical properties of skin and the wound healing process in *Acomys*, *Mus musculus* and *Homo sapiens* is thought to provide useful information for the development of regenerative therapy [4].

Generally, the process of mammalian wound healing is divided into four overlapping phases: haemostasis, inflammation, proliferation, and remodelling [21,22]. During the course of these processes, the mechanical environment at the wound site changes spatiotemporally through the formation of fibrin clots as a temporary extracellular matrix (ECM) and the proliferation and migration of wound healing-related cells such as fibroblasts, myofibroblasts, epidermal keratinocytes and many types of immune cells. Damage to blood vessels causes a rapid haemostatic response and transitions to the inflammatory phase. This process involves vasoconstriction and platelet aggregation. Platelets also act as a source of growth factors and inflammatory cytokines such as transforming growth factor beta-1 (TGF-β1) and platelet derived growth factor (PDGF), promoting fibroblast proliferation and migration to the wound area [23]. Activated fibroblasts exert traction force and induce wound closure (Figure 1b).

In the proliferation phase of *Mus musculus* and *Homo sapiens*, mechanical stress is applied to dermal cells according to the contraction of the wound site, and dermal fibroblasts differentiate into alpha smooth muscle actin (αSMA) positive myofibroblasts, which induce wound site contraction and ECM deposition [24]. However, no fibroblast was observed during the proliferation phase in *Acomys* (Figure 1c) [13].

In the remodelling phase, myofibroblasts are not observed in normal wound healing, whereas proliferative myofibroblasts remain in pathological wound healing and continue to synthesise excess ECM [24]. In fact, it was demonstrated that by applying the same tension experienced by *Homo sapiens* to the skin of *Mus musculus*, the fibroblast-specific focal adhesion kinase (FAK)-extracellular signal-regulated kinase (ERK)-monocyte chemotactic protein 1 (MCP-1) pathway was activated and the number of myofibroblasts increased; as a result, it was possible to reproduce HTS-like pathology in *Mus musculus* [25,26]. In addition, it is known that the mechanical properties of the skin including strength and distribution of tension, which has been identified as the aetiology of keloids, differ by region of the body [27,28,29]. Interestingly, it has been clinically shown that scar formation can be suppressed by relieving tissue tension through procedures such as designing the direction of the wound incision and anxious fixation of wound contraction (Figure 1d) [30,31].

Interestingly, wound-induced hair neogenesis (WIHN), in which new hair follicles develop in the centre of the wound, occurred when a full-thickness wound of sufficient size (>10 mm) was created on the *Mus musculus* [14,15]. A purse string model was applied to understand the wound occlusion associated with the contraction force generated by actomyosin during wound healing [32,33]. In brief, the wound site is stretched and deformed along the direction of tissue scale tension, which is called Langer’s line, and then the wound site is closed by pulling the edges of the wound together with an invisible string [34]. In this model, the edge of the wound is the most tensioned, and if the scar is large enough, the tension is reduced in the centre. Thus, adjusting the wound site to exhibit low tension is thought to help regenerate the complete IOS, including appendages [35].

In summary, tensional homeostasis defined by the physical properties and tension of the tissue plays an important role in the determination of the wound healing modes (i.e., complete regeneration, scar formation, and pathological wound healing) through the regulation of myofibroblast proliferation and differentiation at the wound site via mechanical stress signalling.

### 2.2. Myofibroblast Differentiation and Mechanical Stress Signalling Molecules in the Wound Healing Process

Myofibroblasts are the predominant cell type in granulation tissue of contracting wounds and fibrocontractive diseases such as HTSs and keloids [24,36]. The major function of myofibroblasts is altering tissue tension via force generation by stress fibres and ECM deposition such as Type 1 and Type 3 collagen [37,38].

Myofibroblasts are derived from fibroblasts, fibrocytes, vascular endothelial cells and adipose progenitor cells that have migrated to the wound area. Fibroblasts do not form stress fibres in intact in vivo skin, but differentiate into proto-myofibroblasts under mechanical tension in the wound area. Proto-myofibroblasts form actin-containing stress fibres and secrete ED-A fibronectin. TGF-β1 derived from epidermal keratinocytes, platelets, macrophages, and fibroblasts promotes the secretion of ED-A fibronectin and induces αSMA-positive myofibroblast differentiation [39,40,41,42,43]. TGF-β1 is released from these cells in a biologically latent form (L-TGF-β1) [44]. In the activation of L-TGF-β1, the integrin binds to the prodomain, and exerts a mechanical force to release the active form of TGF-β1 [45]. It is postulated that the increase in myofibroblasts results in a positive feedback loop that accelerates the induction of myofibroblast differentiation at the wound site by increasing tissue tension at the wound site (Figure 2) [23].

In the inflammatory phase, fibroblasts migrate into the wound site. These fibroblasts exert a traction force and close the wound by pulling each other through the ECM. The traction force sensed by integrins and transduced into the biological signalling by mechanotransducer molecules such as FAK, MRTF-A, and YAP/TAZ induces proto-myofibroblast formation. Proto-myofibroblast secretes ED-A fibronectin and activates L-TGF-b. These factors induce myofibroblast differentiation. The traction force exerted by the myofibroblast forms a feedback loop of myofibroblast formation and activation.

Mechanical stress signalling including mechano-sensors, mechano-transducers such as glycocalyx, lipid rafts/caveolin-1, cell adhesion (which is mediated by integrins, hemidesmosomes) and focal adhesion is considered to play important roles in skin structure, function, and pathologies such as HTSs and keloids [46,47]. Among the many mechanical stress molecules that control skin function, the integrin-FAK molecular pathway is the most well-defined regulator of skin mechanotransduction [12,26,48,49,50]. The application of mechanical forces during wound healing leads to the activation of FAK and its downstream pathways, such as phosphatidylinositol 3-kinase (PI3K)/Akt, mitogen-activated protein kinases (MAPK), extracellular-related kinase (ERK), and monocyte chemoattractant protein-1 (MCP-1, also known as CCL2) in fibroblasts and increased HTS formation [51,52,53,54]. Activation of the Rho/ROCK signal promotes cytoskeletal rearrangement, induces the expression of myofibroblast-related genes and myofibroblast differentiation via nuclear translocation of myocardin-related transcription factor A (MRTF-A), and enhances tissue contraction [55]. In contrast, inactivation of the FAK signal causes delayed wound healing, such as diabetic ulcers [49,56,57]. Wingless/Int (Wnt) and yes-associated protein 1 (YAP)/transcriptional coactivator with PDZ-binding motif (TAZ) signalling also activate fibroblast proliferation, myofibroblast differentiation and fibrosis formation via control of TGF-β signalling [58,59,60,61,62,63]. In addition, the low expression of caveolin-1, a cell membrane protein, was involved in cell softening and aberrant responsiveness to mechanical stress in keloid fibroblasts [64]. In summary, tensional homeostasis is a central regulator of skin regeneration because it controls wound healing cell functions, especially myofibroblast functions, through mechanical stress signals.

## 3. Interaction between Immune Response and Mechanical Stress in Wound Healing

### 3.1. Immune Response Regulates Myofibroblast Differentiation and Function

Studies of developmental stage-dependent wound healing patterns suggest that the immune response is an important factor in controlling tissue contraction and fibrosis processes. Human foetuses under 24 weeks gestation have significant regenerative capacity that results in complete recovery of the dermis, epidermis and appendages without scarring [12]. In particular, fibroblast function control by a weak inflammatory response due to the immature immune system plays an important role [65]. Adult wound healing is characterised by a mast cell-mediated influx of neutrophils [66] and macrophages, the secretion of cytokines such as Macrophage Colony Stimulating Factor (CSF-1), tumour necrosis factor (TNF-α), and PDGF to induce differentiation from fibroblasts to myofibroblasts, and the acceleration of fibrogenesis [21,67]. In contrast, foetal wound healing has less invasion and maturation of immune cells such as neutrophils, macrophages, and mast cells, which contributes to a weak inflammatory response and reduced scar tissue formation [68]. Analysis of the essential haematopoietic transcription factor PU.1 null mice lacking macrophages and functioning neutrophils demonstrated a central role of immune cells in scar formation [69]. In these knockout mice, adult skin wounds healed efficiently without scarring. Interestingly, a comparative analysis of gene expression profiles during inflammatory phases in *Mus musculus* and *Acomys* also revealed that the inflammatory response and tissue remodelling in *Acomys* are similar to the foetal wound healing process. Specifically, the expression of inflammatory cytokines such as chemokine (C-X-C motif) ligand 1 (Cxcl1), Cxcl3 Cxcl5, Csf3, and interleukin-1β (IL-1β), which were strongly induced in mice, was not induced in *Acomys*. Alternatively, ECM-related genes such as type 3 and type 5 collagen were upregulated in *Acomys* [70]. These reports clearly show that immune and inflammatory cells play an important role in wound healing fibrosis, but the mechanism is still unclear [3].

Recent studies suggest that crosstalk between fibroblasts and inflammatory cells acts along numerous redundant pathways and is involved in scar formation through the regulation of myofibroblast characteristics and functions [71]. Macrophages, mast cells, neutrophils, and T cells are considered key immune cell types regulating scar formation [72]. Figure 3 summarises the types of immune cells and signal molecules that influence the control of myofibroblast formation and function.

Many studies have confirmed that macrophages play important roles in proper wound healing [73,74]. The most commonly studied macrophage subtypes are referred to as “classically activated” or M1 macrophages and “anti-inflammatory” or M2 macrophages [75]. In the inflammatory phase, M1 macrophages infiltrate the wound area and induce fibroblast proliferation and migration through inflammatory cytokines and growth factors such as interleukin-1 (IL-1), fibroblast growth factor-2 (FGF-2), and PDGF [76]. According to the formation of granulation tissue, microenvironmental cues trigger macrophages to transition into a functionally and phenotypically anti-inflammatory state (M2 macrophages) [77]. M2 macrophages are the main source of TGF-β and induce fibroblast to myofibroblast differentiation [78]. Thus, macrophages regulate fibroblast recruitment and myofibroblast differentiation. Moreover, genetic lineage tracing and flow cytometry revealed the presence of several myofibroblast populations, including CD26-expressing adipocyte precursor cells (APs) and CD29-expressing cells [71]. The ratio of these two types of myofibroblast subpopulations differs in aged skin wounds, cutaneous fibrosis models, and keloids, suggesting their involvement in wound healing modes [79]. Growth factors such as PDGF-C and insulin-like growth factor-1 (IGF-1) secreted by CD301b+ macrophages support the heterogeneity of wound bed myofibroblasts by selectively stimulating cell proliferation of a subset of APs [71]. These reports indicate that further analysis of myofibroblast heterogeneity and subtype functionality is essential for understanding macrophage function in wound healing.

Mast cells are known as immune cells that can induce myofibroblast formation and activity. Platelet-activating factor derived from mast cells induces platelet degranulation and releases many factors, such as TGF-β, PDGF, and fibronectin, all of which stimulate myofibroblast formation. Additionally, mast cells produce a large array of profibrotic cytokines, IL-4, IL-6, IL13-, TNF-a, TGF-β, and PDGF [80], which directly stimulate myofibroblast formation and activities. Neutrophils also secrete various profibrotic cytokines, including TGF-β, IL-6, and vascular endothelial growth factor (VEGF), and may be involved in myofibroblast formation [81]. However, neutrophils are only present in the early stages of wound healing, so they are believed to have a negligible long-term impact [72]. Although T cells are known to play important roles in immune cell recruitment and cytokine secretion, the role of T cells in wound healing is unclear. The T cell subtype TH1 secretes IFN-gamma, IL-2, and TNF-α, while TH2 secretes IL-4, -5, -6, -9, -10, and -13, and many of these are proinflammatory or profibrotic factors [82].

Excessive inflammatory cell recruitment and biofilm formation due to bacterial infections cause chronic wounds. In chronic wounds, the secretion of inflammatory cytokines such as IL-1, TNF-α, and IL-17 by M1 macrophages inhibits the phenotypic shift to M2 macrophages [83]. Furthermore, the increase in MMPs and reactive oxygen species (ROS) derived from neutrophils induces the degradation of growth factors such as ECM and TGF-β [72]. These immune responses may inhibit the formation and activation of myofibroblasts in chronic wounds.

Crosstalk between fibroblasts and inflammatory cells acts along with numerous and redundant pathways, and is involved in scar formation through the regulation of myofibroblast characteristics and functions. Macrophages, mast cells, neutrophils and T cells are typical immune cells involved in the formation and activation of myofibroblasts. The signalling molecules corresponding to each cell are shown in the figure.

In summary, immune cells play an important role in determining the mode of wound healing by controlling fibroblast migration and myofibroblast differentiation through the secretion of inflammatory cytokines and growth factors. However, spatiotemporally regulated cell–cell interactions by various subtypes of cells are extremely complex and require more detailed in vivo analysis and in vitro evaluation for a full understanding.

### 3.2. Mechanical Stress Controls Macrophage Function

It has been suggested that the physical characteristics of tissue and mechanical stress exerted by fibroblasts and myofibroblasts may regulate the functions of immune system cells such as monocytes/macrophages and determine the wound healing mode [84]. During the early phase of wound healing, macrophages migrate to the wound area along the ECM in the inflammatory and proliferative phases, thus sharing their mechanical environment with collagen-producing and contracting cells. In particular, the coordinated activity of macrophages and myofibroblasts plays a key role in controlling normal wound healing, but its perturbation often results in the accumulation and contraction of the ECM known as pathological wound healing [85,86]. Therefore, it is likely that the interaction between macrophages and fibroblasts and/or myofibroblasts is a key regulator in the control of wound healing via the interaction between the immune response and mechanical stress.

Macrophages are the main source of TGF-β, a major regulator of fibroblast function in the inflammatory phase [78]. TGF-β-mediated intercellular crosstalk is established only in the very close range from secretory cells. Fibroblasts and myofibroblasts need to be close enough to crosstalk with macrophages, but the attractive mechanism is unknown. As candidate controlling mechanisms, chemotaxis and ECM cues such as traction force, stiffness, or topographies are known [87,88,89]. In particular, the traction force exerted by fibroblasts and myofibroblasts creates mechanical cues that can be sensed far beyond chemotactic gradients (100–200 μm) by cells sharing the same substrates [90,91]. Traction force exerted by fibroblasts generates deformation fields in the fibrillar collagen matrix that provide far-reaching mechanical cues for macrophages. Integrin α2β1 and stretch-activated channels mediate macrophage migration (Figure 4) [17].

Integrins regulate macrophage functions such as proliferation, differentiation, migration, and phagocytosis through not only various signal transductions, but also cell–cell and cell–ECM adhesion. Several types of integrins such as α1β2, αMβ2, αxβ2, α4β1, α4β7, and αvβ3 are expressed on the cell surface of macrophages, and can bind to ECMs such as collagen, fibronectin, and fibrinogen. Extrinsic mechanical stress via ECM and intrinsic traction force exerted by the cytoskeleton can change the conformation of integrin molecules, which activates mechanical stress signalling such as FAK and ERK signalling. From this, macrophages may detect ambient tension via integrins and regulate cell function [92]. Periodic stretch stimulation, an in vitro mechanobiological evaluation system, promotes the secretion of IL-6, IL-8, and TNF-α in human alveolar macrophages, human monocyte-derived macrophages, and THP-1 cells [93]. In THP-1 cells, cyclic stretching induces cyclooxygenase (COX)-2 gene expression, which stimulates the production of prostaglandin E 2 (PGE2) [94]. Both tension and ECM stiffness may control macrophage function. It has been reported that adhesion and cytokine secretion of macrophages differentiated from human THP-1 cells are dependent on matrix stiffness [95]. In studies using polyethylene glycol (PEG)-coated polyacrylamide gels with stiffness in the range of 1.4–348 kPa, macrophages tended to adhere to harder matrices, with TNFα secretion maximal at 1.4 kPa [95].

These reports suggest that tensional homeostasis and the feedback loop of the immune response play important roles in the deterioration of the myofibroblast-mediated fibrotic response, but further elucidation of the detailed mechanism needs to be considered.

## 4. In Vitro Functional Analysis Model of Mechanical Stress

The results from in vivo studies comparing wound healing in different animal species, ages, and body sites suggest that tensional homeostasis and immune responses play important roles in wound healing and tissue reconstruction. Due to clinical ethical limitations, it is extremely difficult to analyse the functionality and molecular function of tissue-scale tensional homeostasis for adult human skin physiology, including wound healing. Conventional in vitro mechanobiology studies mainly use silicon culture vessels and cellular and molecular biological approaches to analyse the effects of the stiffness of the cell adhesion surface, mechanical elongation and compression stimulation on planar cultured cells (Figure 5) [96,97]. However, in wound healing studies that require consideration of the spatiotemporal regulation of cellular function by mechanobiological factors such as ECM orientation, strength, and cell–cell interactions with skin and immune system cells, the information provided by the 2D culture experimental system is extremely limited (Figure 5).

Dermal equivalents composed of collagen gel containing fibroblasts are used as a model that can evaluate ECM remodelling and functional control of fibroblasts by tissue-scale tension [98]. By modifying the culture environment of the dermal equivalents (floating, tethered collagen gel, stretching, etc.), the mechanical environment of fibroblasts can be regulated. The mechanical tension load on the dermal equivalents includes not only genes related to myofibroblast formation such as αSMA and vinculin, but also matrix synthesis genes such as collagen, tenascin, hyaluronan, plasminogen activator inhibitor (PAI)-1, 2, and tissue inhibitor of metalloproteinase (TIMP) [98,99]. This phenotype is similar to the migration and proliferation response of fibroblasts found in wound healing. These phenotypic decisions are controlled by mechanical stress signalling molecules such as integrins, FAK signalling, and MRTF-A [98,100]. Dermal equivalents are also useful for analysing cell- and tissue-scale morphological changes due to the traction force exerted by fibroblasts [101]. By embedding the dermal equivalents in a cell-free collagen gel, it was shown that the traction force generated at the interface between the dermis and collagen gel is functional for tissue contraction [102]. This report supports the biomechanical interpretation of the pulse string model in wound closure. Coculture experiments with fibroblasts and macrophages in collagen gel revealed that ECM remodelling with fibroblasts provides mechanical cues and controls macrophage migration (see Section 3.2 for details) [17].

It has been reported that human skin equivalent (HSE) can better reproduce skin physiological functions such as cell–cell interaction, cell differentiation, tissue formation function, and drug absorption and metabolism than the 2D culture model [103,104,105]. HSE is considered a useful model for the analysis of skin physiological function and drug discovery [106,107]. In the field of wound healing research, it has been applied as an evaluation model for the inflammatory response, re-epithelialisation response, and contraction response of wounds [108,109]. Recently, HSE has been applied to the safety evaluation of healthcare products, enabling sustainable research and development through a reduction in animal experiments [106].

Currently, one of the common HSEs in the field of basic research is Bell’s model, which combines a dermal equivalent consisting of collagen gel containing fibroblasts and an epidermal keratinocyte sheet. Since the traditional Bell’s model induces tissue contraction due to suspension culture in the process of constructing a dermal equivalent [103,110], there is no tensional homeostasis in the living body, and it is not suitable for functional analysis of mechanical stress under physiological conditions (Figure 5c). Therefore, we have developed a tensional homeostatic skin model (THS model) that reproduces tensional homeostasis by mechanically fixing the tissue to a culture insert (Figure 5d) [18]. The THS model showed the same skin marker protein expression and skin barrier function as natural skin and general skin equivalents [18]. As a result of detailed histological analysis, the orientation of fibroblasts and ECM in the tension direction, which is a characteristic of natural skin, was observed in the THS model, but the orientation was lost in Bell’s model [18]. Furthermore, as a result of skin functionality analysis, it was suggested that tensional homeostasis promotes epithelial turnover through epithelial–mesenchymal interaction by keratinocyte growth factor (KGF) [18]. In the THS model, it was confirmed that the ECM synthesis function was improved by the change in the mechano-sensing state due to the increase in integrin α2 expression and the activation of the mechanical stress signal by the nuclear translocation of MRTF-A [18]. Interestingly, the THS model showed improved reactivity to retinoic acid and vitamin C derivatives [18]. These results clearly show that the control of physiological function depends on the presence or absence of tension in the HSEs and demonstrate that the THS model is a useful tool for evaluating skin physiological function.

## 5. Future Perspective

The progress made in the past decade has been remarkable, paving the way for possible future wound healing induction without HTS or keloid formation. Studies on tissue morphology, functional control by tensional homeostasis and subtype analysis of myofibroblast lineages provide evidence of traditional scar formation suppression approaches through skin incision and suturing along the Langer line. Furthermore, because inflammatory and immune responses control scar formation through the control of myofibroblast subtypes and functionality, there is a possibility of developing a skin reconstruction method that combines immune response control and tissue tension control. Additionally, by elucidating the wound healing mechanism of the cell–cell interaction between myofibroblasts, the immune system and hair follicles through tensional homeostasis in adult human skin, it is expected that a complete reconstruction method can be developed for skin and appendages, including hair follicles and sweat glands. One of the major concerns in the development of treatments for adult human wounds, inspired by species- and developmental stage-dependent tissue remodelling mechanisms, is the absence of an appropriate in vitro model applicable to evaluate cell–cell interactions in human wounds and drug efficacy. HSE, an in vitro model capable of assessing tensional homeostasis cell–cell interactions, tissue morphology, skin function, and drug response, is a suitable research model for wound healing, and previous studies have clearly demonstrated the mechanisms of re-epithelialisation by epithelial–mesenchymal interactions. In recent years, tissue engineering, organoids, and organ-on-chip approaches have been applied to develop next-generation HSEs that can evaluate interactions with immune cells, blood vessels, and hair follicles [111,112,113,114,115]. In the next 10 years, the HSE will be updated to a skin organ system equivalent model, and it will be possible to evaluate biological physiological functions such as mechanical stress and the immune response, providing a new approach to realise skin regeneration [16].

## Figures and Tables

**Figure 1 ijms-22-05474-f001:**
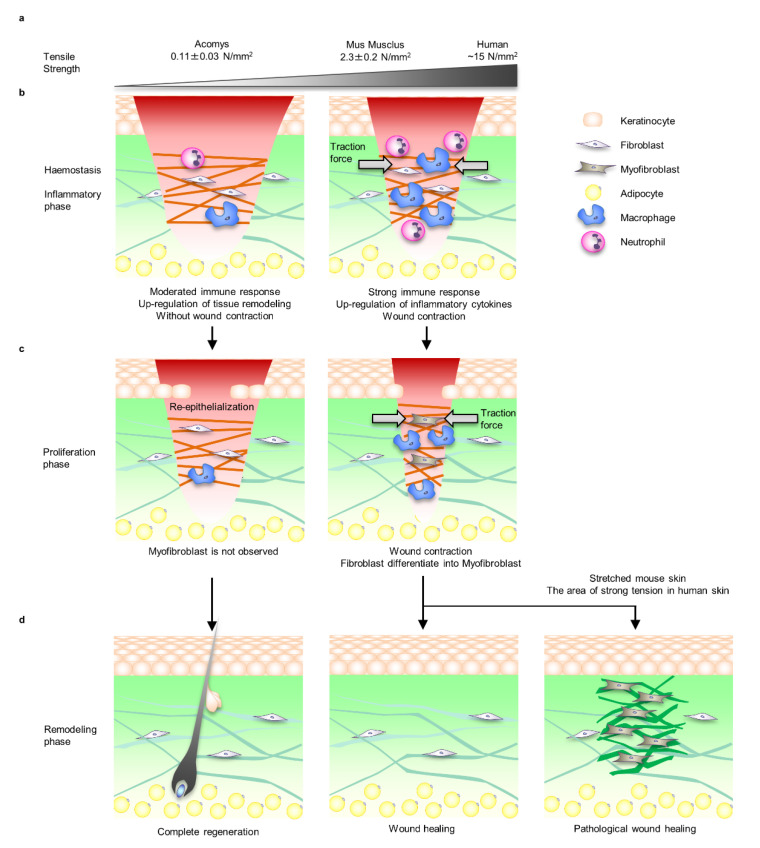
Comparison of *Acomys*, *Mus musculus*, and *Homo sapiens* skin tensile strength and wound healing process. (**a**) The skin of *Acomys* is particularly soft and the tensile strength is 1/20 that of mice. On the other hand, the tensile strength of human skin, which fluctuates depending on the site and direction, is approximately 5 times that of *Mus musculus*. (**b**) During the inflammatory phase of *Mus musculus* and *Homo sapiens*, immune cells such as macrophages and neutrophils infiltrate along the fibrin matrix of the fibrin clot to protect against bacterial infections and remove debris. At this time, inflammatory cytokines and chemokines are secreted, and fibroblasts begin to proliferate and migrate into the wound area. *Mus musculus* and *Homo sapiens* fibroblasts that migrate to the fibrin clot exert tensile force and close the wound. On the other hand, in *Acomys* wounds, wound contraction was not observed, the inflammatory response was mild, and the reorganisation responses of ECMs such as collagens and matrix metalloproteinases (MMPs’) gene expression were upregulated. (**c**) During the proliferative phase, the fibrin clot is replaced with highly vascularised granulation tissue containing fibroblasts and macrophages. Fibroblasts differentiate into myofibroblasts via mechanical stress signals due to the tension loading associated with wound closure. Myofibroblasts’ proliferation and differentiation, wound contraction and collagen fibre deposition become upregulated. No myofibroblasts are observed during the proliferative phase in *Acomys*. (**d**) During the remodelling phase, which lasts up to a year, scar formation progresses by ECM synthesis, degradation, and cross-linking. Myofibroblasts gradually decrease during this phase, but it is unclear whether the mechanism is due to apoptosis or differentiation into other cells such as fibroblasts and adipocytes. Wounds in *Mus musculus* and *Homo sapiens* do not heal to the same level mechanical properties as intact skin. In *Homo sapiens* skin with relatively strong tensile strength, myofibroblasts often remain and pathological wound healing such as HTS and keloid healing occurs. Interestingly, it has been reported that HTS is induced in *Mus musculus* by applying the same level of tension as humans. *Acomys* completely regenerates the skin organ system including hair follicles.

**Figure 2 ijms-22-05474-f002:**
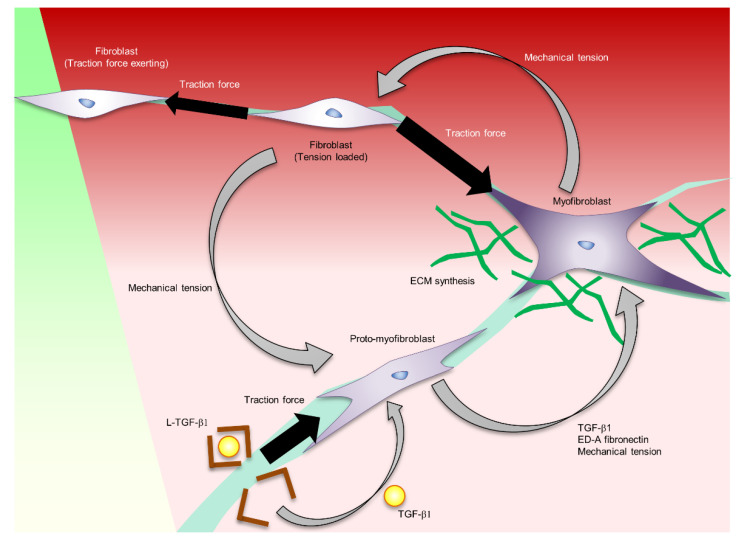
The regulation of myofibroblast formation and function controlled by mechanical stress.

**Figure 3 ijms-22-05474-f003:**
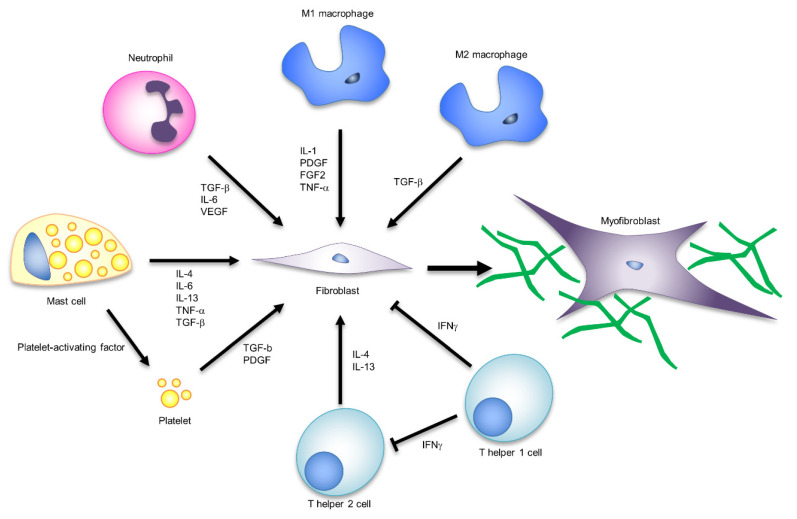
The influence of immune responses on myofibroblast formation and function.

**Figure 4 ijms-22-05474-f004:**
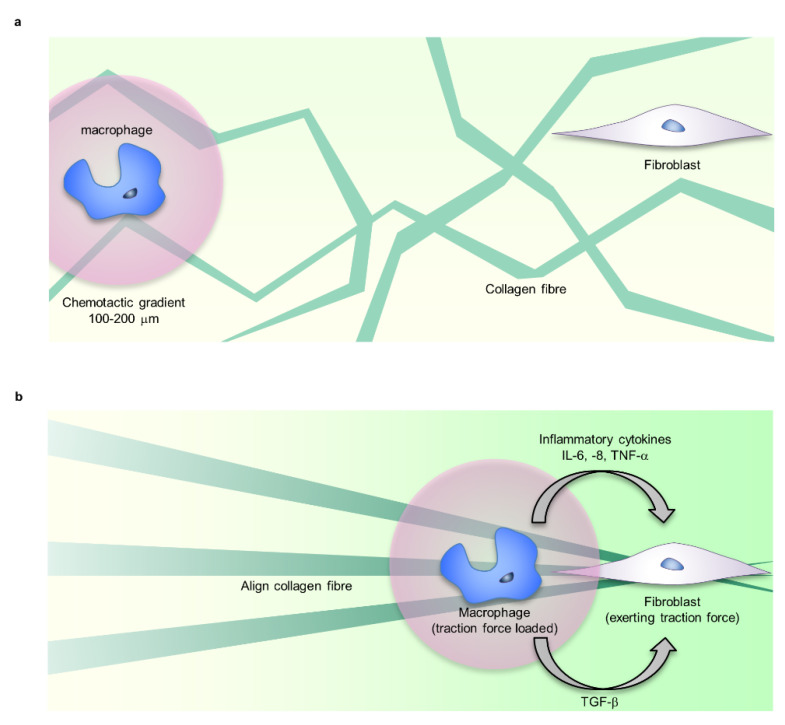
Mechanical stress controls macrophage function. (**a**) Fibroblasts and macrophages share ECM, such as fibrin, fibronectin, and collagen as adhesive scaffolds. (**b**) The traction force exerted by fibroblasts forms the alignment and tension distribution of ECM fibre. Macrophages sense the traction force via ECM far beyond chemotactic gradients (100–200 μm), migrate to the vicinity of fibroblasts, and crosstalk via TGF-b. Additionally, changes in ECM mechanical properties regulate the inflammatory response of macrophages. The mechanical regulation of macrophage function is involved in inducing myofibroblast formation.

**Figure 5 ijms-22-05474-f005:**
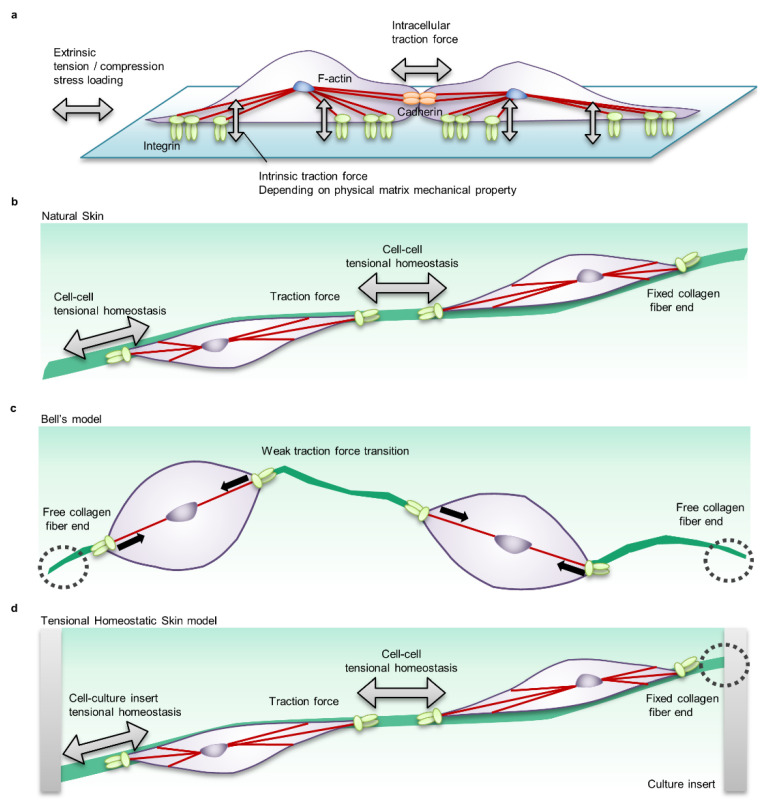
Cell-scale mechanical interactions in in vitro mechanobiology research models. (**a**) Cell–cell mechanical interactions in a common two-dimensional planar cell culture system. In this system, various stiff materials such as conventional tissue culture plastic (nonphysiological condition, ~2 GPa), polymer substrates (fibrosis condition, 50–100 kPa), and silicone (physiologically soft condition, 2–5 kPa) are selected as the matrix to evaluate cellular, morphological and functional regulation by the mechanical condition. Cells adhere to the matrix via adhesive proteins such as integrin receptors and form intercellular junctions via membrane proteins such as cadherin. The actin cytoskeleton transmits the intrinsic traction force that cells generate to the matrix and adjacent cells, determining cell morphology, motility, and functionality. In addition, a system that stretches or loosens the tension of the matrix is used to evaluate the effects of extrinsic mechanical loading. (**b**) Mechanical interaction in natural skin. Fibroblasts in natural skin adhere to ECM such as Type 1 and Type 3 collagen fibres that are three-dimensionally oriented. The actomyosin-mediated cellular traction force is transmitted to neighbouring cells via ECM fibres. The state in which the intrinsic cellular traction force is balanced via the ECM is called tensional homeostasis. (**c**) Mechanical interaction in the common skin equivalent (Bell’s model). The dermal equivalent in Bell’s model consists of fibroblasts suspended in collagen gel. Since Bell’s model undergoes floating culture during tissue maturation and does not fix collagen fibre ends, the traction force of fibroblasts causes tissue contraction and the tensional homeostasis level is adjusted to a low level. (**d**) Mechanical interaction of the tensional homeostatic skin (THS) model. In the THS model in which the tissue ends of the dermis are clamped in a culture insert, the traction force of fibroblasts is efficiently transmitted to neighbouring cells, and tensional homeostasis is reproduced.

## Data Availability

Not applicable.

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
