# Peer review of "Mechanical and Immunological Regulation in Wound Healing and Skin Reconstruction"

_ijms, 2021, doi:10.3390/ijms22115474_

Round 1

Reviewer 1 Report

The authors present a review on mechanical and immunologiacal regulation in wound healing and skin reconstruction. While the abstract sounds promising, the following text does not meet these expectations. While reading, some questions remain unanswered:
- why do the authors focus on murine skin?
- what about the role of other important key factors such as MMPs?
-
why are the skin models not explained in more detail, e.g. in a Figure?

Overall, the review touches on several topics without going into detail and without really reflecting the entire scientific status. In addition, the review needs some revision with regard to the language and sources. Some examples:
- lines 244-248: W
hat references does this refer to?
- lines 272-273: How do the authors know that Bell´s model is most commonly used?
- lines 279-280: Where is this data shown?
- line 280 and following: where can the reader find the histological analysis? No references are given in this passage.

Author Response

Response to Reviewer 1 Comments

Comments and Suggestions for Authors

The authors present a review on mechanical and immunologiacal regulation in wound healing and skin reconstruction. While the abstract sounds promising, the following text does not meet these expectations. While reading, some questions remain unanswered:

Our Response: We are grateful for your evaluation and valuable suggestions for our manuscript. Wound healing is a very complex process. Many analyses have been reported from the respective areas of immunology and mechanobiology. Although, there are several interesting reports on the interaction between the two, it has not reached its comprehensive understanding. The purpose of this review was to integrate information and discuss the potential of skin equivalents as a new research model. Considering your suggestion, we modified the structure of the manuscript and added several topics.  Our specific responses are listed below:

Comments 1. why do the authors focus on murine skin?

Our Response: Comparative analysis of wound healing mechanisms between several species is expected to provide insights into the key regulators of scar-free wound healing (Harn, H. et. Al., Exp. Dermatol 2019). In particular, complete skin regeneration and/or scar-free wound healing reported in some specific species (e.g., Acomys, Mus musculus, and Homo sapiencs) and in the biophysical environment (e.g., stretched skin, and large wound) are expected to provide important information for human therapeutic development (Ito, M. et. al., Nature, 2007, Seifert, A. et. Al., Nature, 2012). We have modified the structure of the manuscript in introduction and section 2.1. In the new version, to prevent information confusion, we described the differences in mammalian regeneration mechanisms along the wound healing process sequence.

Comments 2. what about the role of other important key factors such as MMPs?

Our Response: We are grateful for your suggestion to strengthen for our study. We investigated the latest report on the role of mechanical stress and immune cell interactions in chronic wounds. We added in Section 3.2 that neutrophil and macrophage type-dependent effects on myofibroblast formation and functional regulation via MMPs (line 349-355).

Comments 3. why are the skin models not explained in more detail, e.g. in a Figure?

Our Response: In accordance this helpful comment, we have added an interesting topic on fibroblast-macrophage interactions via growth factors and mechanical stress which was revealed by using a three-dimensional culture model of fibroblasts in collagen gel (this technique is substantially homologous to skin equivalents). This topic provided the latest and deeper insights into the control of macrophage motility and function by mechanical stress in a three-dimensional environment. Figure 5 aims to compare the biomechanical environment of in vitro research models useful for the analysis of mechanical stress (line 445-463).

Comments 4. Overall, the review touches on several topics without going into detail and without really reflecting the entire scientific status. In addition, the review needs some revision with regard to the language and sources. Some examples:

lines 244-248: What references does this refer to?

Our Response: In accordance this helpful comment, we have added reference information (line 422-426).

Comments 5. lines 272-273: How do the authors know that Bell´s model is most commonly used?

Our Response: In several reports and reviews on skin equivalents, construction method of Bell’s model is introduced as a typical construction method (Ali, N. et. Al., Br J Dermatol, 2015, Kimura. S. et. Al., Commun Biol. 2020). However, other types of HSEs such as decellularized dermis are used for skin grafting in the medical field. Therefore, we have revised the manuscript more accurately (line 473-475).

Comments 6. lines 279-280: Where is this data shown?

Our Response: In accordance this helpful comment, we have added reference information (line 473-494).

Comments 7. line 280 and following: where can the reader find the histological analysis? No references are given in this passage.

Our Response: In accordance this helpful comment, we have added reference information (line 473-494).

We hope that these changes meet with your approval.  We greatly appreciate your comments, which provided a helpful perspective on our work.

Reviewer 2 Report

A significant stream of research in biomedicine is the use of new methods of microscopy, tissue engineering and biomechanical modeling in order to broaden the knowledge about the mechanisms of damaged tissues repair. This research is highly desirable and has a long-term significance. Currently, some results have already been applied in the field of regenerative medicine and tissue engineering. The MS presented for evaluation is therefore up-to-date and contains interesting content from the scope mentioned above.

However, some adjustments and additions may be suggested to make this review more "friendly" to the potential reader, including those for whom this topic is not the main area of interest.

Comments

- The introduction in many parts of the text of comparisons of wound healing mechanisms in amphibians, teleosts, mice (Acomys) and humans causes a certain information chaos, which makes it difficult to perceive the most important data. The comprehensive description of Fig. 1. appears to be largely satisfactory for this purpose.

- It may be suggested that the authors try to find a better balance between mechanobiology and immunology processes. In the current version, there is a distinct advantage in descriptions of physical phenomena, at the expense of clear descriptions of immunoregulation of wound healing and complications of this process.

- Presumably, it would be very helpful to compile a graphical or tabular summary of the information contained in the text of subsections 2.2 and 3.1; 3.2.

- A much deeper analysis of the in vitro models used in mechanobiology was expected than described in Section 4, including the model developed by the authors of this review. Of course, you can refer to the authors' earlier publication (no. 85), but a bit more comment here would be useful.

- It is a pity that the authors did not "touch on" (at least in a short comment) the problem of disturbances in the regulation of wound healing in the case of infection (acute and chronic conditions). Most often then, there is excessive inflammation, abnormal wound closure, scarring, keloid formation, etc. How does mechanical stress signaling work in such cases? How then is the interaction between the response of the immune system and the mechanical stress. Addressing the above-mentioned problems would increase the substantive value of this study.

- It may be suggested to quote interesting information contained in several publications, not covered by the authors: Tottoli EM et al. Pharmaceutics, 2020; El Ayadi A et al. Int J Molec. Sci. 2020; Larouche J et al. Adv Wound Care, 2018...

- It is advisable to correct editorial and linguistic errors.

Author Response

Response to Reviewer 2 Comments

Comments and Suggestions for Authors

A significant stream of research in biomedicine is the use of new methods of microscopy, tissue engineering and biomechanical modeling in order to broaden the knowledge about the mechanisms of damaged tissues repair. This research is highly desirable and has a long-term significance. Currently, some results have already been applied in the field of regenerative medicine and tissue engineering. The MS presented for evaluation is therefore up-to-date and contains interesting content from the scope mentioned above.

However, some adjustments and additions may be suggested to make this review more "friendly" to the potential reader, including those for whom this topic is not the main area of interest.

Our Response: We have studied your comments carefully and found that you understood the value and significance of our study in this field.  We are grateful for your evaluation and valuable suggestions for our manuscript. Our specific responses are listed below:

Comments 1. The introduction in many parts of the text of comparisons of wound healing mechanisms in amphibians, teleost, mice (Acomys) and humans causes a certain information chaos, which makes it difficult to perceive the most important data. The comprehensive description of Fig. 1. appears to be largely satisfactory for this purpose.

Our Response: We are grateful to you for this suggestion. We have revised the structure of text in introduction and Section 2.1. In the new version, to prevent information confusion, we have described the differences in mammalian wound healing mechanisms along the wound healing process.

Comments 2. It may be suggested that the authors try to find a better balance between mechanobiology and immunology processes. In the current version, there is a distinct advantage in descriptions of physical phenomena, at the expense of clear descriptions of immunoregulation of wound healing and complications of this process.

Our Response: In accordance this helpful comment, we have added several topics in Section 3.1 regarding the control mechanisms of myofibroblast formation and function, by several immune cell.

Comments 3. Presumably, it would be very helpful to compile a graphical or tabular summary of the information contained in the text of subsections 2.2 and 3.1; 3.2.

Our Response: In accordance with your comment, we added new figures corresponding to section 2.2, 3.1, and 3.2 (Fig. 2, 3, 4).

Comments 4. A much deeper analysis of the in vitro models used in mechanobiology was expected than described in Section 4, including the model developed by the authors of this review. Of course, you can refer to the authors' earlier publication (no. 85), but a bit more comment here would be useful.

Our Response: In accordance this helpful comment, we have added an interesting topic on fibroblast-macrophage interactions via growth factors and mechanical stress which was revealed by using a three-dimensional culture model of fibroblasts in collagen gel (this technique is substantially homologous to skin equivalents). This topic provided the latest and deeper insights into the control of macrophage motility and function by mechanical stress in a three-dimensional environment (line 375-394, line 445-463).

Comments 5. It is a pity that the authors did not "touch on" (at least in a short comment) the problem of disturbances in the regulation of wound healing in the case of infection (acute and chronic conditions). Most often then, there is excessive inflammation, abnormal wound closure, scarring, keloid formation, etc. How does mechanical stress signaling work in such cases? How then is the interaction between the response of the immune system and the mechanical stress. Addressing the above-mentioned problems would increase the substantive value of this study.

Our Response: We are grateful for your suggestion to strengthen for our study. We investigated the latest report on the role of mechanical stress and immune cell interactions in chronic wounds. We added in Section 3.1 that macrophage type-dependent effects on myofibroblast formation and functional regulation, however, the exact mechanism remains unclear (line 349-355). The effects of immune response and mechanical stress on chronic wounds that you have pointed out are important issues in this research area.

Comments 6. It may be suggested to quote interesting information contained in several publications, not covered by the authors: Tottoli EM et al. Pharmaceutics, 2020; El Ayadi A et al. Int J Molec. Sci. 2020; Larouche J et al. Adv Wound Care, 2018...

Our Response: Thank you for kindly sharing the citation information. These reports were very helpful in this revision and have been included in the reference.

Comments 7. It is advisable to correct editorial and linguistic errors.

Our Response: Our manuscript has been proofread by the English editing service.

We hope that these changes meet with your approval.  We greatly appreciate your comments, which provided a helpful perspective on our work.

Reviewer 3 Report

1. lines 35-37: "An incomplete tissue reconstruction response causes abnormal skin function and, perceived as ulceration, fibrosis, pigmentation, wrinkles and sagging, and poor quality of life." --> An incomplete tissue reconstruction response causes abnormal skin function perceived as ulceration, fibrosis, pigmentation, wrinkles and sagging, and poor quality of life.

2. line 58: formations --> formation

3. lime 66: Define α-SMA.

4. line 88; " ... ECM ..." --> extracellular matrix (ECM)

5. lines 159 & 251: Italicize in vivo.

6. line 161: " ... TGF-b ..." --> TGF-beta or TGF-β

7. line 213: " ... population ..." --> populations

8. line 256: Italicize in vitro.

9. line 285: KGF --> Keratinocyte growth factor.

Author Response

Response to Reviewer 3 Comments

Comments and Suggestions for Authors

  1. lines 35-37: "An incomplete tissue reconstruction response causes abnormal skin function and, perceived as ulceration, fibrosis, pigmentation, wrinkles and sagging, and poor quality of life." --> An incomplete tissue reconstruction response causes abnormal skin function perceived as ulceration, fibrosis, pigmentation, wrinkles and sagging, and poor quality of life.

  1. line 58: formations --> formation

  1. lime 66: Define α-SMA.

  1. line 88; " ... ECM ..." --> extracellular matrix (ECM)

  1. lines 159 & 251: Italicize in vivo.

  1. line 161: " ... TGF-b ..." --> TGF-beta or TGF-β

  1. line 213: " ... population ..." --> populations

  1. line 256: Italicize in vitro.

  1. line 285: KGF --> Keratinocyte growth factor.

Our Response: We have studied your comments carefully and found that you understood the value and significance of our study in this field. We are grateful for your evaluation and valuable suggestions for our manuscript. We have collected what you pointed out in our manuscripts. Additionally, our manuscript has been proofread by the English editing service.

We hope that these changes meet with your approval.  We greatly appreciate your comments, which provided a helpful perspective on our work.

Reviewer 4 Report

This is a good review providing an overview about scarless wound healing process focusing on the potential involved mechanobiological and immunological aspects which would influenced the wound healing mode. 

I just would ask for a language revison. 

Author Response

Response to Reviewer 4 Comments

Comments and Suggestions for Authors

This is a good review providing an overview about scarless wound healing process focusing on the potential involved mechanobiological and immunological aspects which would influenced the wound healing mode. 

I just would ask for a language revison. 

Our Response: We have studied your comments carefully and found that you understood the value and significance of our study in this field. We are grateful for your evaluation and valuable suggestions for our manuscript. Our manuscript has been proofread by the English editing service.

We hope that these changes meet with your approval.  We greatly appreciate your comments, which provided a helpful perspective on our work.

Round 2

Reviewer 1 Report

I appreciate the authors' revisions.Nevertheless, the text does not meet the expectations of the abstract from my point of view.
In the abstract it should be mentioned that different species are compared. The manuscript should be improved again to address linguistic deficiencies.  

Author Response

Response to Reviewer 1 Comments

I appreciate the authors' revisions. Nevertheless, the text does not meet the expectations of the abstract from my point of view.
In the abstract it should be mentioned that different species are compared. The manuscript should be improved again to address linguistic deficiencies.  

Our Response: We appreciate your careful review and comments. We added to the abstract that this review is based on a comparison of wound healing mechanisms between different species and developmental stage. In addition, our manuscript has been checked by the English proofreading service.

We hope that these changes meet with your approval.  We greatly appreciate your comments, which provided a helpful perspective on our work.